# Physical Activation of Wooden Chips and the Effect of Particle Size, Initial Humidity, and Acetic Acid Extraction on the Properties of Activated Carbons

**Davide Bergna** [1,2] , **Henrik Romar** [1,2,*] **and Ulla Lassi** [1,2]

1   Research Unit of Sustainable Chemistry, University of Oulu, P.O. Box 3000, FI-90014 Oulu, Finland; davide.bergna@chydenius.fi (D.B.); ulla.lassi@oulu.fi (U.L.)
2   Unit of Applied Chemistry, Kokkola University Consortium Chydenius-University of Jyvaskyla, Talonpojankatu 2B, FI-67100 Kokkola, Finland
*   Correspondence: henrik.romar@chydenius.fi; Tel.: +358-50-358-2939

**Abstract:** In this research study, two different wooden biomasses (birch and pine) were thermally carbonized and steam-activated into activated carbons in a one-stage process. The effects of particle size and humidity (as received and oven-dried) on the properties, such as specific surface areas, pore volumes, and pore size distributions, of the final activated carbon characteristics were examined. Another set of biomasses (birch, spruce, and pine) was pre-treated before carbonization and the activation steps through an extractive process using a weak acetic acid in Soxhlet extractors. According to the results, the dried samples had a slightly lower surface area, while no difference was observed in the yields. For the extracted samples, there was a significant difference, especially in the pore size distributions, compared to the non-extracted samples. There appeared to be a shift from a meso-microporous distribution to a microporous distribution caused by the extractive pre-treatment.

**Keywords:** biomass; carbonization; activation; activated carbon; acetic acid; pore distribution

---

## 1. Introduction

Activated carbons (ACs) are used for several purification applications in different industrial processes, including wastewater treatment, gas cleaning processes, and metal removal from waste streams [1–3]. AC has also been used as support for heterogeneous catalysts [4–6] or as a catalyst itself [7,8]. The consumption and potential applications of AC are continuously increasing globally [9].

Most carbon-containing substances, including industrial waste fractions, can be carbonized through thermochemical processes into substances with a high carbon content. These substances can be further converted into ACs through chemical or physical activation processes, including thermal treatment steps.

Traditionally, ACs have been prepared from a number of carbon-rich biomasses, such as coconut shells, coal (lignite), and sawdust, as raw materials [2,10,11]. The thermal conversion of biomass into activated carbon is a two-step process beginning with a carbonization phase performed in an inert atmosphere. In this first step, the biomass is converted into solid carbon. This carbonization occurs under an inert gas atmosphere (i.e., nitrogen) at temperatures ranging from 873 to 1073 K [11]. The product from the initial carbonization step contains a high level of carbon (up to 60%), but metals and ashes are also present, which are concentrated from the initial biomass during this process.

In the second step, the carbon is activated chemically or physically, creating structures with well-defined pore size distributions depending on the final use of the AC. Most biomass-based materials with significant carbon content can be steam-activated through a two-stage process

following the reactions listed below [12]. Reaction (1) describes a summary of the carbonization step, and Reactions (2)–(5) are reactions during the steam activation step.

$$C_x(H_2O)_y \rightarrow xC_{(s)} + yH_2O \tag{1}$$

$$C + H_2O \rightarrow CO + H_2 \quad \Delta H = 118.9 \text{ kJ mol}^{-1} \tag{2}$$

$$CO + H_2O \rightarrow CO_2 + H_2 \quad \Delta H = -40.9 \text{ kJ mol}^{-1} \tag{3}$$

$$C + CO_2 \rightarrow 2CO \quad \Delta H = 159.7 \text{ kJ mol}^{-1} \tag{4}$$

$$C + 2H_2 \rightarrow CH_4 \quad \Delta H = -87.4 \text{ kJ mol}^{-1} \tag{5}$$

During the chemical activation process, the carbonized biomass is first impregnated with activating agents, such as phosphoric acid, zinc chloride sodium, or potassium hydroxides, and then it is dried and activated at temperatures between 670 and 870 K in an inert atmosphere [13]. The activating agents in the carbons are removed by an acid wash followed by cleanup with distilled water and then dried and characterized. One of the advantages of chemical activation is the production of more uniform pore distribution and higher carbon yields.

A typical chemical activation mechanism involves KOH because of the capability of $K^+$ ions at high temperatures (700–1225 K) to form intercalating compounds with carbon and $K_2CO_3$ that thermally decompose releasing $CO_2$ that creates the pores [14]:

$$KOH + C \rightarrow K_2CO_3 + K_2O + 2H_2 \tag{6}$$

$$K_2CO_3 + C \rightarrow K_2O + 2CO \tag{7}$$

$$K_2CO_3 \rightarrow K_2O + CO_2 \tag{8}$$

$$2K + CO_2 \rightarrow K_2O + CO \tag{9}$$

There is extensive, ongoing research to identify new raw materials that can be converted into AC. New sources that can be used include fractions regarded as industrial waste, such as lignin from pulping processes and even used car tires [15–17]. One problem with these materials is that they can contain rather high levels of metals and other inorganics, resulting in an ash content in the final products (particularly acid water-soluble ashes) higher than the level that can be accepted by end-users of ACs [18], and the quality demands are highly dependent on the final use. Most of the ashes are not soluble in water at a pH level close to neutral, but they can easily dissolve if the carbons are used at low pH conditions, eventually contaminating the process in which they are used.

A number of methods have been applied to decrease the amount of ashes accumulated during the carbonization and activation processes. The demineralization process can be applied directly to the biomass used, to the intermediate formed during carbonization, or to the final product. To minimize the ash content, different chemicals have been tested. In one study, rice straw was demineralized using a number of chemicals, including distilled water, acetic acid, and low concentrations (5%) of hydrochloric, sulfuric, nitric, and orthophosphoric acids [19]. Carbon black prepared from used car tires has been demineralized using sulfuric acid in different concentrations [20].

This study was performed to investigate how the properties (particle size and humidity) of the starting materials affect the final products. During the study, a series of wood-based biomasses were carbonized and steam-activated in a one-stage process. Four different sizes of woodchips were used (2 mm, 6 mm, 2 cm, sawdust), and the effects of particle size, wood species, moisture content, and extraction pretreatment were examined to determine yields, carbon content, and specific surface areas. The extraction was performed using Soxhlet extraction with a weak organic acid (acetic acid). The extraction process was performed to identify a potential pre-treatment method to improve the properties of the final products.

## 2. Materials and Methods

### 2.1. Biomasses Used for Carbonization and Activation

Wooden chips from birch (*Betula pendula*) and spruce (*Picea abies*) were used for the carbonization and activation experiments. The chips were sieved to three different sizes: 2 mm, 6 mm, and 2 cm. During carbonization and activation, the chips were processed as received (Ar) and dried (D) for 24 h at 378 K. For the extraction tests (Ex), the sawdust (Sw) of birch (*B. pendula*), spruce (*P. abies*), and pine (*Pinus sylvestris*) were used. Table 1 presents the sample nomenclature.

**Table 1.** Sample nomenclature.

| Sample | Description |
|--------|-------------|
| Ar | Carbonized and activated as received |
| D | Oven-dried before carbonization and activation |
| Sw | Sawdust activated untreated |
| Ex | Sawdust acetic acid-extracted and activated |

### 2.2. Determination of Moisture Content

Known masses of the chips were weighted into tared beakers and dried at 378 K until reaching a constant weight. Moisture content, given as a percentage, was calculated as the loss of mass divided by the initial mass.

### 2.3. Carbonization and Activation Procedure

The carbonization and activation of the different particle size samples were performed in a single-stage process using a rotating quartz reactor inserted in a Nabertherm oven. About 400 g of each sample was inserted into the reactor, and during the carbonization step, the temperature ramped from room temperature to 1073 K (10 K/min). At 1073 K, the temperature was maintained for 2 h, and during this period, the reactor was flushed with 120 g/h of overheated steam (413 K) produced by a Bronkhorst steam generator that was fed with a constant volume of water using nitrogen gas as a carrier. During the entire process, the reactor was flushed with nitrogen to prevent the oxidation of the samples. The oven was cooled overnight and continuously flushed with nitrogen. A schematic presentation of the experimental setup is shown in Figure 1.

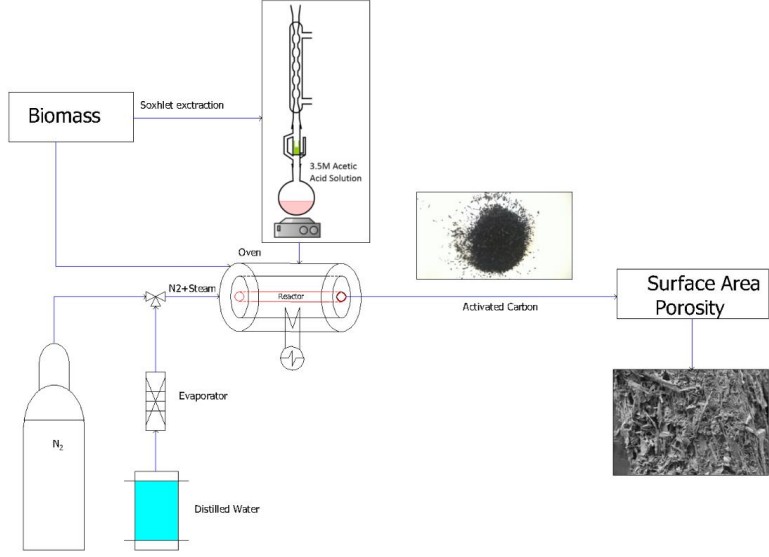

**Figure 1.** Scheme and reactor setup for extraction, carbonization, and activation.

Because of the small amounts of extracted samples available, the carbonization and activation of the sawdust samples were performed as a one-stage process in a fixed-bed, stainless-steel reactor (Φ 12 mm) inserted in a tubular oven (Nabertherm GmbH, Lilienthal, Germany). The parameters used for carbonization and activation were the same, with the exception that the mass of the samples was 5 g/sample, and the steam feed 30 g/h.

*2.4. Determination of Yields and Total Carbon Content*

The mass yield from each sample was calculated as the mass of wood-based AC divided by the mass of the initial sample used for carbonization and activation.

The content of carbon present in the initial chips (dried) used and in the ACs produced, given as total carbon (TC) percent, was measured using a solid-phase carbon analyzer (Skalar Primacs MCS, Breda, Netherlands). Dried and crushed samples were weighted in quartz crucibles and combusted at 1373 K in an atmosphere of pure oxygen, and the $CO_2$ formed was analyzed using an IR analyzer (Skalar Formacs, Breda, Netherlands). Carbon content values were obtained by reading the signal of the IR analyzer from a calibration curve derived from known masses of a standard substance, i.e., citric acid. The total mass of carbon in each sample was calculated as a percent of the mass initially weighted.

*2.5. Specific Surface Area and Pore Size Distribution*

Prior to the measurements, portions of each sample (about 200 mg) were pretreated at low pressures and high temperatures to clean the surfaces. Adsorption isotherms were obtained by immersing the sample tubes in liquid nitrogen (76 K) to obtain isothermal conditions. Nitrogen was added to the samples in small steps, and the resulting isotherms were obtained. Specific surface areas were calculated from adsorption isotherms according to the Brunauer–Emmett–Teller (BET) method [21]. Pore size distributions were calculated using the Density functional theory (DFT) algorithm assuming a slit geometry of the pores. The t-Plot model was used considering Harkins and Jura thickness curves and a Langmuir surface area with a correlation coefficient higher than 0.999. Due to the setup of the instrument used, the Micromeritics ASAP 2020, pores as small as 1.5 nm in diameter can be measured.

*2.6. Extraction Process*

A 4 h long demineralization process was performed using 3.5 M acetic acid (Sigma Aldrich, Saint Louis, MO, USA) as an extracting agent. Aliquots of each wood species were transferred into the cellulose thimbles of the Soxhlet extraction unit. A sawdust size of less than 2 mm was selected for extraction. After demineralization, the samples were suction-filtered on glass filters and washed with distilled water until the pH of the filtrates was neutral. Finally, the samples were dried for 24 h at 378 K in an oven equipped with mechanical convection. The samples were finally carbonized and steam-activated according to the procedure described in Section 2.3. The carbons produced were analyzed for specific surface areas and pore size distributions, as described in Section 2.5.

## 3. Results and Discussion

The biomass fractions from different wood species used in this study can be regarded as waste fractions. These biomasses are also known to have a rather high initial level of carbon and can easily be converted into activated carbon.

*3.1. Effect of Particle Size on AC Porosity*

Table 2 reports the values of yield, humidity, and total carbon divided by different particle sizes and initial moisture content. The results show a small difference between wood species and the particle sizes considered. In particular, the yield for the process parameters selected was about 5%. On a dry base, this value increased to 10–15% proportionally with the initial moisture content (around 50%).

**Table 2.** Total carbon content, humidity, and overall yield after activation.

| Sample | Total Carbon Content (%) | Yield (%) | Humidity (%) |
| --- | --- | --- | --- |
| Spruce_2 cm_Ar | 71.1 | 5 | 50 |
| Spruce 6 mm_Ar | 94.5 | 4.9 | - |
| Spruce_2 cm_D | - | 12 | - |
| Spruce_2 mm_Ar | - | 5 | - |
| Birch_2 cm_Ar | 87.6 | 4 | 52 |
| Birch_6 mm_Ar | 89.9 | 2.5 | - |
| Birch_2 cm_D | - | 14 | - |
| Birch_2 mm_Ar | - | 4.5 | - |

In Table 3, the porosity characteristics of the different particle sizes are reported. In this case, a small reduction in the BET surface is more evident for the pre-dried samples. For the pore distribution from both the DFT model and the t-Plot, no evident change was detected, and there was an almost equal distribution between micropores and mesopores. As can be observed, the wood type also has a small influence on porosity, suggesting that the main factor is correlated with the activation process parameters. It is evident that steam activation, as known, favors the creation of mesoporosity. The different particle sizes in the range considered for this experiment did not affect the porosity characteristics, indicating that the activation process is a surface reaction, which suggests that optimizing the surface-to-volume ratio in a way other than only the size can be a potential option to improve the activated material. For the pre-dried samples, no relevant difference was observed in the porosity of the final AC.

*3.2. Effect of the Extraction Process on AC Porosity*

During thermal steam activation, some mass was removed from the precursors, which was wood sawdust in this case. The remaining mass obtained after drying the activated samples was calculated as the percent of the initial mass and the denoted yield. In general, the yield after the activation step was higher for the extracted samples compared to the direct activated samples at 14–17% and 3–10%, respectively. This difference could have been caused by the removal of moist content as well as by the fact that easily removable structures were already extracted during the pretreatment step. The same structures were most likely removed also during the thermal activation of the untreated precursors. Eventual mass losses during the extraction step were not measured in this study.

Although the specific surfaces were rather similar, there was an evident difference regarding the pore size distributions. This was visible because of a shift in the shape of the adsorption isotherms, as illustrated in Figure 2.

The isotherms obtained from the untreated samples were all Type II, indicating the presence of mesopores. Isotherms obtained from the extracted samples before carbonization and activation were all Type I, indicating that most of the pores were in the micropore region.

The results from the analysis of specific surface areas (BET model) and pore size distributions (DFT model) showed that the steam-activated samples had specific surface areas exceeding 1000 $m^2$/g, as shown in Table 4. This applied to both untreated activated and extracted activated samples.

**Table 3.** Specific surface areas (SSA) and pore size distributions of the different activated carbons (ACs). BET: Brunauer–Emmett–Teller; DFT: Density functional theory.

| Calculation Method | Unit | Spruce 2 cm_Ar | Spruce 2 cm_D | Spruce 6 mm_Ar | Spruce 2 mm_Ar | Birch 2 cm_Ar | Birch 2 cm_D | Birch 6 mm_Ar | Birch 2 mm_Ar |
|---|---|---|---|---|---|---|---|---|---|
| BET | | | | | | | | | |
| SSA | $m^2/g$ | 1066 | 983 | 1048 | 990 | 906 | 833 | 1063 | 1029 |
| Pore volume | $cm^3/g$ | 0.848 | 0.710 | 0.800 | 0.700 | 0.625 | 0.590 | 0.765 | 0.740 |
| C value | | 1438 | 2229 | 1646 | 2783 | 2702 | 2371 | 934 | 3123 |
| Langmuir surface area (c.corr. > 0.999) | $m^2/g$ | 1323 | 1217 | 1287 | 1227 | 1121 | 1040 | 1280 | 1262 |
| t-plot | | | | | | | | | |
| micropore volume | $cm^3/g$ | 0.221 | 0.208 | 0.205 | 0.210 | 0.186 | 0.165 | 0.225 | 0.199 |
| micropore area | $m^2/g$ | 499 | 398 | 496 | 509 | 364 | 322 | 551 | 472 |
| External surface area | $m^2/g$ | 555 | 505 | 576 | 498 | 472 | 473 | 538 | 575 |
| DFT | | | | | | | | | |
| pore volume | $cm^3/g$ | 0.736 | 0.627 | 0.689 | 0.623 | 0.535 | 0.544 | 0.656 | 0.647 |
| µpores | $cm^3/g$ | 0.289 | 0.276 | 0.289 | 0.279 | 0.256 | 0.236 | 0.298 | 0.282 |
| mesopores | $cm^3/g$ | 0.446 | 0.348 | 0.397 | 0.343 | 0.27 | 0.305 | 0.355 | 0.355 |
| macropores | $cm^3/g$ | 0.001 | 0.003 | 0.003 | 0.001 | 0.009 | 0.003 | 0.003 | 0.010 |
| µpores | % | 39 | 44 | 42 | 45 | 48 | 43 | 45 | 44 |
| mesopores | % | 61 | 56 | 58 | 55 | 50 | 56 | 54 | 55 |
| macropores | % | 0 | 0 | 0 | 0 | 2 | 1 | 1 | 1 |

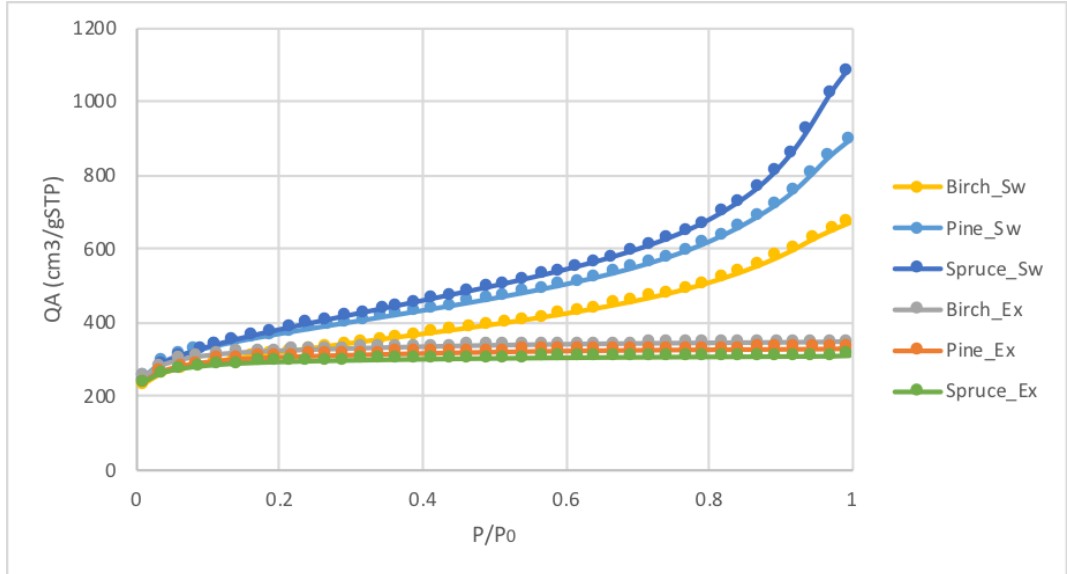

**Figure 2.** Isotherm comparison between unextracted and acetic acid-extracted wood-based AC.

**Table 4.** Porosity characteristics of the ACs produced from unextracted and extracted sawdust.

| Calculation Method | Unit | Spruce_Sw | Birch_Sw | Pine_Sw | Spruce_Ex | Birch_Ex | Pine_Ex |
|---|---|---|---|---|---|---|---|
| BET | | | | | | | |
| SSA | $m^2/g$ | 1302 | 1190 | 1296 | 1125 | 1109 | 1141 |
| Pore volume | $cm^3/g$ | 1.669 | 1.053 | 1.260 | 0.538 | 0.476 | 0.511 |
| C value | | 3611 | 1415 | 1657 | 1095 | 3812 | 4043 |
| Langmuir surface area (c.corr. > 0.999) | $m^2/g$ | 1407 | 1393 | 1464 | 1530 | 1335 | 1462 |
| t-plot | | | | | | | |
| micropore volume | $cm^3/g$ | 0.090 | 0.160 | 0.138 | 0.397 | 0.321 | 0.396 |
| micropore area | $m^2/g$ | 155 | 346 | 275 | 993 | 824 | 990 |
| External surface area | $m^2/g$ | 1147 | 843 | 1020 | 231 | 284 | 151 |
| DFT | | | | | | | |
| pore volume | $cm^3/g$ | 1.370 | 0.973 | 1.280 | 0.455 | 0.395 | 0.427 |
| μpores | $cm^3/g$ | 0.290 | 0.310 | 0.330 | 0.408 | 0.376 | 0.392 |
| Mesopores | $cm^3/g$ | 1.050 | 0.640 | 0.900 | 0.046 | 0.016 | 0.033 |
| Macropores | $cm^3/g$ | 0.030 | 0.023 | 0.05 | 0.001 | 0.003 | 0.002 |
| μpores | % | 21 | 32 | 26 | 90 | 95 | 92 |
| Mesopores | % | 76 | 66 | 70 | 10 | 4 | 7 |
| Macropores | % | 2 | 2 | 4 | 0 | 1 | 1 |

The change in pore volume distribution can also be observed from the calculations of pore volumes according to the DFT algorithm. Particularly, the results showed that the extraction process increased the micropore percentage in the activated samples, while the untreated activated samples had a predominant presence of mesopores. For untreated and activated wood sample biomass, the micropore portion was around 30%, while the portion of micropores in the extracted samples was around 90%, as calculated from the pore volumes.

The shift in pore distribution to micropores in the extracted samples compared to the untreated samples can most likely be explained as a decrease in the content of the metals needed to create mesopores during the thermal activation process. These findings are also an indication that the reaction of steam with the biomass (physical activation) is the most important factor in the production of ACs but not the only factor; therefore, it appears that some degree of chemical activation occurs.

Table 4 reports the comparisons between untreated sawdust samples and the extracted samples run in the same reactor and under the same process conditions.

The total carbon content of the pretreated samples, as shown in Table 5, was slightly lower compared to the untreated samples, which was probably due to the previous extraction of organic compounds in the starting material that "compensated" for the ash reduction. A further investigation

not part of this study could involve determining how acetic acid differently affects the biomass structure during extraction and during the thermochemical decomposition process.

**Table 5.** Total carbon content and yield of extracted sawdust-based AC.

| Sample | Total Carbon Content (%) | Yield (%) |
| --- | --- | --- |
| Spruce_Ex | 89.5 | 14 |
| Birch_Ex | 87.8 | 17 |
| Pine_Ex | 84.1 | 12 |

There was a significant difference in the pore volumes, as shown in Tables 3 and 4, even if the same species of wood and the same carbonization and activation parameters were used. There were two major differences between the series. One was the shape of the biomass used. The data shown in Table 3 were produced using woodchips made in a cutting mill, producing needle-shaped chips in a fiber direction, while the data shown in Table 4 were produced from the sawdust of fibers that were cut in a direction perpendicular to the wood fibers. The other major difference was the oven used. For the Table 3 data, a rotating quartz oven was used, while for the Table 4 data, a fixed-bed reactor made of stainless steel was used.

## 4. Conclusions

It is well known that a number of parameters affect the properties of ACs produced from biomass. The results achieved in this study indicate that different particle sizes of the starting biomass have a small but measurable influence on AC porosity. A previously dried biomass appears to produce ACs with a slightly lower BET surface area. The pre-extraction of the biomass using acetic acid induces a significant increase in the microporosity during steam activation. This change in pore size distribution is most likely due to the lack of chemical activating agents in the form of metals and minerals naturally present in the wood. To some extent, the change could be a result of alterations in biomass composition caused by the extraction process. The use of a pre-extraction process with the resulting change in pore size distribution represents a versatile method to produce ACs with specific pore size distributions without affecting the specific surface areas.

**Author Contributions:** Conceptualization, D.B. and H.R.; Methodology, D.B., H.R., U.L.; Investigation, D.B.; Writing-Original Draft Preparation, H.R., D.B.; Writing-Review & Editing, D.B., H.R.; Supervision, U.L., H.R.

**Funding:** The study was supported by and performed within the project Renepro (20200224) funded by Interreg Nord (H.R.). Economical support was received from the Central Ostrobothnia Cultural Foundation (D.B.).

**Conflicts of Interest:** The authors declare no conflict of interest.

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
