# Peer review of "Physical Activation of Wooden Chips and the Effect of Particle Size, Initial Humidity, and Acetic Acid Extraction on the Properties of Activated Carbons"

_carbon_

Round 1
Reviewer 1 Report
The studies presented in the reviewed manuscript “Physical activation of wooden chips, the effect of particle size, initial humidity and acetic acid extraction on the properties of activated carbons” reads interesting. However, I recommend a significant revision to check for grammar, ambiguity, and inappropriate sentences. I have provided a few examples below; I recommend the authors to look for similar corrections before submission of the revision.
1. Line 17, change week to weak
2. Line 57, What is meant by “capability of K at high”? what K stands for?
3. Line 81, Change “orto-phosphoric acid” to Orthophosphoric acid
4. Line 98, Change “as mass lost” to mass of loss
5. How feasible is this technique? Is it possible to scale-up this process?
6. Are these samples crystalline or amorphous?
7. What can be done using these activated carbons? Can this material used for any applications?
Reviewer 2 Report
The manuscript by Bergna et al. showed the effect of particle size, moisture content, and solvent extraction for characteristics of activated carbon. Although there are interesting results, it is difficult to follow because the purpose of this study have not been clarified. For example, which one point is the author want to clarify, good pretreatment method, the effects of acid treatement, difference in tree species? In short, the intent of authors is confusing and these materials are messy. In addition, there are too many mistakes in the paper. It is just like a manuscript draft. Line 148-149 and Line 152-153 are exactly the same sentences. Do you explain abbreviations of diagrams in main text or footnotes? Why are references numbered twice? I strongly encourage the authors to carefully correct the manuscript and reconsider the design of the paper.
Round 2
Reviewer 1 Report
The authors included recommended suggestions. In light of this, I recommend acceptance of this paper for further consideration.
Reviewer 2 Report
Since the authors are revised the paper, I think this research is suitable for publication in C.